# Dual-Enzyme Cascade Composed of Chitosan Coated FeS_2_ Nanozyme and Glucose Oxidase for Sensitive Glucose Detection

**DOI:** 10.3390/molecules28031357

**Published:** 2023-01-31

**Authors:** Bowen Shen, Molan Qing, Liying Zhu, Yuxian Wang, Ling Jiang

**Affiliations:** 1State Key Laboratory of Materials-Oriented Chemical Engineering, College of Biotechnology and Pharmaceutical Engineering, Nanjing 211816, China; 2School of Chemistry and Molecular Engineering, Nanjing Tech University, Nanjing 210009, China; 3College of Food Science and Light Industry, Nanjing Tech University, Nanjing 210009, China

**Keywords:** nanozyme, immobilized enzyme, double-enzyme cascade, glucose detection

## Abstract

Immobilizing enzymes with nanozymes to catalyze cascade reactions overcomes many of the shortcomings of biological enzymes in industrial manufacturing. In the study, glucose oxidases were covalently bound to FeS_2_ nanozymes as immobilization carriers while chitosan encapsulation increased the activity and stability of the immobilized enzymes. The immobilized enzymes exhibited a 10% greater increase in catalytic efficiency than the free enzymes while also being more stable and catalytically active in environments with an alkaline pH of 9.0 and a high temperature of 100 °C. Additionally, the FeS_2_ nanozyme-driven double-enzyme cascade reaction showed high glucose selectivity, even in the presence of lactose, dopamine, and uric acid, with a limit of detection (LOD) (S/N = 3) as low as 1.9 × 10^−6^ M. This research demonstrates that nanozymes may be employed as ideal carriers for biological enzymes and that the nanozymes can catalyze cascade reactions together with natural enzymes, offering new insights into interactions between natural and synthetic biosystems.

## 1. Introduction

As with mostly protein-based biological catalysts, enzymes accelerate chemical reactions by reducing activation energy [1]. Due to their great substrate specificity and high catalytic efficiency, natural enzymes have significant uses in biochemical analysis, medicine, chemical synthesis, and food preparation [2,3,4,5]. However, the usage of enzymes is costly, their recycling is challenging, and their tertiary protein structure is easily damaged under harsh process conditions, resulting in the loss of catalytic activity [6,7,8]. Additionally, the stability of enzymes must meet strict standards in the industrial manufacturing environment, greatly restricting the broader industrial application of enzymes to date [9,10,11].

Accordingly, enhancing enzyme stability in harsh environments is a good strategy to promote the use of enzymes [12,13]. Enzyme immobilization by encasing them in solid substances or restricting them to certain regions using physical or chemical methods enables their reuse while also greatly increasing their stability and catalytic efficiency [14,15,16,17]. Additionally, the immobilization of enzymes makes it easier to separate enzymes from products, culture medium, and substrates, which improves process economics [18,19]. Conventional enzyme-immobilization carriers are chemically inert and have no catalytic effect themselves [14,20,21]. Some carrier materials even limit the contact between the substrate and the enzyme, thereby affecting the catalytic efficiency. Additionally, certain carriers will interact with the enzyme to reduce its ability to catalyze reactions [22,23,24]. Therefore, the most important factor for successful enzyme immobilization is the selection of an appropriate carrier material [25].

The creation of new and highly effective nanozymes is another strategy for addressing the shortcomings of enzymes [26,27]. Since in 2007 it was discovered that Fe_3_O_4_ has enzyme-like capabilities, nanozymes have gained significant attention [28]. Nanozymes are frequently employed in medicine and sensing because of their benefits for the easy combination of active sites, high stability, and simple synthesis [29,30,31]. Although it has been noted that nanozymes can imitate the activities of oxidases, peroxidases, and hydrolases, little is known about how nanozymes interact with other enzymes [26]. As an upgraded version of the typical Fe_3_O_4_ nanozyme, iron sulfide (FeS_2_) nanozymes have excellent enzyme-like catalytic activity, mimic the properties of other enzymes, and have abundant active sites such as amino and sulfhydryl groups [32,33]. Moreover, many active groups of FeS_2_ are ideal conjugation sites for enzyme immobilization, while their defect-prone structures and active surface groups promote non-covalent interactions with natural enzymes [34,35]. The scope of catalytic reactions can theoretically be greatly expanded by combining enzymatic processes and nanozymes, enabling the implementation of advanced cascade reactions. Chemical reactions are also facilitated by electron transport on the nanozyme surface [36,37].

In our previous research, iron ions and cysteine were used to create FeS_2_ nanozymes, which showed good peroxidase-like activity [38]. Due to its durability, chemical inertness, and flexibility, chitosan is an ideal polysaccharide material that may be employed as a carrier for enzyme reactions. We constructed an immobilized enzyme system based on chitosan (CS) encapsulation and covalent binding of nanozymes, demonstrating a high glucose oxidase (Gox) loading capacity and enhanced Gox stability in harsh environments. Notably, the presence of nanozymes enhanced the catalytic activity and stability of Gox. In addition, biological enzymes and nanozymes can support cascade reactions that catalyze the production of hydroxyl radicals from glucose, making it possible to detect glucose in urine.

## 2. Results and Discussion

### 2.1. Characterization of FeS_2_/CS@Gox

In order to assess the state of the immobilized enzyme, the morphology of the FeS_2_ enzyme particles was first studied. The FeS_2_ nanomaterial exhibited a homogeneous hexagonal nanosheet shape (Appendix A), whereas the FeS_2_ with immobilized enzymes had a hexagonal nanosheet shape with compound surface attachments (Figure 1A). When Gox was loaded onto the surface of FeS_2_ nanosheets, polymeric attachments were clearly visible (Figure 1B). The enzyme is physically and chemically bonded to the surface of FeS_2_, preserving the FeS_2_’s original form in the composite material. The establishment of the link between FeS_2_ and Gox was then confirmed by FTIR (Figure 1C and Appendix A), and the typical absorption peak of cysteine-rich FeS_2_ nanosheets was observed at 3300–3400 cm^−1^ which was attributed to the NH stretching vibration. The symmetric and asymmetric stretching vibrations of CH and CH_2_ were reflected in the peaks at 2750 and 3200 cm^−1^ [39]. The presence of S-H groups in FeS_2_ was confirmed by the weak signal at 2550 cm^−1^. Stretching vibrations of the C=O molecule resulted in both asymmetric and symmetric peaks at 1600–1700 cm^−1^. The broad peak at 900–1300 cm^−1^ was attributed to C=S double bonds, while the broad peak at 1600–1700 cm^−1^ can be attributed to the newly formed C=O of NH_2_ in FeS_2_ and COOH in Gox after conjugation with CS and Gox. The N-H bending vibration caused a peak at 1538 cm^−1^, while the vibration peaks at 1295 and 1059 were related to the C-O and C-N stretching vibrations, respectively. The distinctive infrared functional group peaks demonstrate the chemical interaction between FeS_2_ and Gox. The successful conjugation was further confirmed by the elemental distribution in Fe/CS@Gox (Appendix A). Finally, pure FeS_2_ displayed distinct crystal characteristic peaks, which also demonstrated that FeS_2_ had been successfully prepared. The same crystal characteristic peaks also developed in FeS_2_ when the enzyme was immobilized. The consistency of the XRD pattern also demonstrated that Gox and CS did not alter the structure of FeS_2_ after immobilization (Figure 1D). FeS_2_ can efficiently exhibit its catalytic activity provided that its crystal structure remains unaffected, which is advantageous for the creation of immobilized enzyme systems. These results confirmed that Gox was successfully grafted onto the FeS_2_ nanosheets.

### 2.2. Immobilization Ratio and Immobilized Enzyme Activity

The conjugation of Gox to the FeS_2_ nanosheets was confirmed by green fluorescent labeling. As shown in Appendix A, there was a clearly visible green fluorescence under ultraviolet light which confirmed the presence of Gox in the FeS_2_/CS particles. Furthermore, fluorescence microscopy of the FeS_2_/CS@Gox particles showed that FeS_2_ exhibited an aggregated morphology in the field of view, and bright green fluorescence coincided with the distribution of FeS_2_ (Figure 2A,B). Subsequently, we investigated the effect of adding different amounts of Gox to the enzyme immobilization ratio. The enzyme immobilization did not exceed 30% when only FeS_2_ and Gox were linked covalently, but the FeS_2_/CS had higher immobilization efficiency. FeS_2_/CS had the highest efficiency and loading capacity when Gox was added at 0.5 mg, which was the most suitable concentration for enzyme immobilization in this system (Appendix A and Figure 2C). In an aqueous solution, Gox readily self-dissociates when it is directly chemically bound to FeS_2_, but CS creates a confined space for FeS_2_@Gox, making Gox dissociation challenging. The effects of pH and temperature on immobilized enzymes were then further investigated. As shown in Figure 2D,E, the immobilized enzyme exhibited higher catalytic activity than the free enzyme, which may be due to the attachment to the support changing the functional groups of the enzyme. In addition, it has been reported that the sulfide helps to enhance the activity of the enzyme, and the catalytic activity of Gox can be significantly improved by the numerous thiol groups in FeS_2_ [40]. Additionally, the immobilized enzymes demonstrated improved catalytic activity in extreme circumstances. The immobilized enzyme may retain more than 80% of its catalytic activity in acidic and alkaline environments, whereas the catalytic activity of the free enzyme is less than 60%. The catalytic activity of enzymes is also impacted by temperature. Free enzymes barely exhibit any catalytic activity at 100 °C, but immobilized enzymes exhibit more than 60% catalytic activity. The results demonstrated that the stability and catalytic activity of the enzyme were significantly improved by immobilizing it with FeS_2_ and CS. Reusability is a crucial aspect of immobilized enzymes, and Figure 2F shows the reuse of the FeS_2_/CS@Gox. The catalytic efficiency of FeS_2_/CS@Gox remained above 80% over the first eight reuse cycles, but further usage eventually resulted in structural degradation and a loss of mass transfer, which would lower catalytic performance [41]. The use of enzyme engineering in actual manufacturing will be encouraged by its excellent reusability, which will lower costs and boost yield.

### 2.3. The POD-like Activity of FeS_2_

Subsequently, we investigated the peroxidase-like catalytic capacity of the composite through the oxidation of 3,3′,5,5′-tetramethylbenzidine (TMB) in the presence of H_2_O_2_. We tested the effects of nFeS concentration, H_2_O_2_ concentration, pH, and temperature on the peroxidase-like activity. With the increase of FeS_2_ concentration, the oxidation rate of TMB also increased (Figure 3B). Moreover, with the increase of H_2_O_2_ concentration, the rate of TMB oxidation also gradually increased (Figure 3C). After optimizing the temperature and pH for the peroxidase-like activity of nFeS, it was found that the peroxidase-like activity of FeS_2_ decreased with the increase in temperature, and the optimum pH was 4.0 (Figure 3D,E). The above results show that the peroxidase-like catalytic performance of FeS_2_ nanoparticles is dependent on catalyst concentration, substrate concentration, temperature, and pH, similar to natural enzymes. The mechanism of peroxidase-like activity of FeS_2_ nanoparticles was then further explored. Terephthalic acid was used as a fluorescent probe to detect the production of hydroxyl radicals, which can be produced from H_2_O_2_ by peroxidase [42]. The generation of hydroxyl radicals from H_2_O_2_ catalyzed by FeS_2_ was detected. In the presence of H_2_O_2_, FeS_2_ can convert TA into TA-OH, which has increased fluorescence intensity. The production of hydroxyl radicals was detected under acidic or neutral conditions, indicating that the peroxidase-like activity of FeS_2_ is due to its ability to decompose H_2_O_2_ to produce hydroxyl radicals (Appendix A). To determine the catalytic efficiency of FeS_2_ nanosheets, we performed enzyme kinetic analysis with testing at optimal pH and room temperature. Unexpectedly, FeS_2_ nanosheets showed a very high affinity for H_2_O_2_, with a *K_m_* of 0.01743 mM and a *V_max_* of 110.9848 nM/s (Figure 3F). The results demonstrate that FeS_2_ mimics natural peroxidase catalytically and has the benefit of stability over natural enzymes. The excellent peroxidase-like activity of FeS_2_ nanosheets encouraged us to further apply them for two-enzyme coupled catalysis.

### 2.4. Sensitivity and Selectivity of Glucose Detection

Based on the exceptional peroxidase-mimicking activity of FeS_2_, we initially employed the immobilized Gox-TMB system to detect glucose in a solution. With the increase of glucose content, the absorbance at 652 nm of the detecting system also increased (Figure 4A). The color response of the immobilized Gox-TMB system showed a good linear relationship with the glucose concentration (Figure 4B). Moreover, the detection limit was as low as 1.9 × 10^−6^ M, which was sufficiently sensitive for the actual glucose sample. We created a standard color card with the colors representing the various glucose concentrations in order to more easily determine the concentration of glucose in the solution (Figure 4C). The selectivity was assessed by examining the reactions to the typical interfering substances, including the selectivity of FeS_2_ for H_2_O_2_ and the selectivity of the system for glucose. At the same substrate concentration, only H_2_O_2_ produced the response of FeS_2_, while there was no reaction with common metal ions and TMB, demonstrating its strong selectivity for H_2_O_2_ (Appendix A). Similarly, only glucose caused a discernible change in solution color, while the solutions containing interfering compounds remained colorless and behaved exactly like the blank, clearly demonstrating the great selectivity for glucose (Figure 4D and Appendix A). The FeS_2_/CS@Gox system’s strong selectivity and sensitivity for glucose make it suited for the identification of samples that contain the substance. In addition, the catalytic efficiency of the glucose detection system based on FeS_2_/CS@Gox was further evaluated over time. As shown in Figure 4E, the catalytic activity of the glucose detection system decreased during one month of storage, but the overall value was higher than 80%, demonstrating the good storage stability of the glucose detection system. The good shelf life of the FeS_2_/CS@Gox system will also further improve its application in the detection of real samples. The nanozyme-based glucose-sensing system offers a good detection limit [43], and our system offers obvious benefits over other systems (Table 1). In conclusion, the FeS_2_/CS@Gox system constructed by the immobilized enzyme strategy has excellent sensitivity, selectivity, and shelf life and is a potential product for glucose detection.

### 2.5. Glucose Detection in Actual Samples

Patients suffering from kidney disease or diabetes have high levels of glucose in their urine, and their physical condition can be judged by detecting the glucose content in urine or blood [48,49]. To assess the viability of FeS_2_/CS@Gox, we created a mixed solution that contained glucose to mimic actual samples. The standard color card provides a rough estimate of the concentration of glucose when the test sample reacts with the detection system (Figure 5A). The glucose concentration in the sample to be tested was further judged by measuring the absorbance, and the glucose concentration that was calculated using the standard curve is shown in Table 2 and Figure 5B. When we compared the detected concentration to the actual concentration of the material to be tested, there was only a slight discrepancy, and the error was less than 3%. This finding demonstrates that the glucose detection system based on FeS_2_/CS@Gox can determine the concentration of glucose present in different complex samples, such as blood and urine. In addition, we can also detect glucose in blood and sweat using the FeS_2_/CS@Gox system, and the stability based on the FeS_2_/CS@Gox system can be applied to the development and production of glucose detection medical devices.

## 3. Materials and Methods

### 3.1. Materials

The reagents were purchased from commercial sources and were used without further purification. Ferric chloride (FeCl_3_) was purchased from Macklin (Shanghai, China). Cysteine and 3,3′,5,5′-tetramethylbenzidine (TMB) were purchased from Sigma-Aldrich (Shanghai, China). Chitosan (CS) and Glucose oxidase (Gox) were purchased from Sangon Biotech (Shanghai, China). DNS Reducing Sugar Detection Kits were purchased from Solarbio (Beijing, China).

### 3.2. Synthesis and Characterization of FeS_2_

FeS_2_ was synthesized as described previously, with minor modifications as follow [38]. Briefly, 0.5 g of FeCl_3_ and 1.0 g of cysteine were dissolved in 40 mL of ethylene glycol, and then 3.6 g sodium acetate was added. The mixture was then allowed to react for 12 h at 200 °C in an autoclave. The produced black powder was collected, washed with ethanol three times, and dried at 60 °C.

### 3.3. Immobilization of Gox

In order to promote covalent conjugation, Gox and FeS_2_ were mixed in 0.1 mM phosphate buffer with pH = 9 and reacted at 37 °C for 24 h under constant stirring at 12,000 rpm. The solids were then separated, added to the CS solution without washing, and left for 10 min to allow chitosan to coat the alkaline FeS_2_ surface. FeS_2_ particles containing Gox and CS were centrifuged, washed three times with distilled water, and lyophilized. The loading of the FeS_2_ particles with the enzyme was confirmed by the FITC fluorescent labeling and confocal laser scanning microscopy (Nikon, Tokyo, Japan) [50].

### 3.4. Materials Characterization

The morphology of the prepared FeS_2_ and FeS_2_/CS@Gox was observed using a scanning electron microscope (SEM, S4800). The solid sample was dispersed in absolute ethanol, dried on a silicon wafer, scanned, and photographed using SEM. A transmission electron microscope (TEM) was used to further analyze the morphology of FeS_2_/CS@Gox. FeS_2_/CS@Gox was dispersed in an ethanol solution, dried on a copper grid, and then examined and photographed under the TEM. The dried product was positioned on a quartz glass slide for X-ray diffraction examination (XRD) (Rigaku Miniflex 600, Tokyo, Japan). With 0.008 θ steps and a 3-degree-per-minute scanning speed, the scanning range was 5 to 60 degrees. In order to prepare the FeS_2_ and FeS_2_/CS@Gox powders for Fourier transform infrared spectroscopy, they were combined with dry KBr powder in an agate mortar, crushed to a fine powder, and then heated to remove any leftover moisture. After being formed into a pellet, the combined powder was scanned using the device. For the Fourier transform infrared spectrum of the Gox solution, KBr was first made into thin slices, and then a small amount of Gox solution was dropped on the KBr thin slices and scanned by the instrument.

### 3.5. The Effect of pH on the Activity of Immobilized Enzyme

To study the effect of pH on free enzymes and immobilized enzymes, the free enzymes and immobilized enzymes were put in different pH reaction systems to catalyze glucose, the enzyme activities under different conditions were recorded, and the highest activity of the free enzymes was defined as 100% relative enzyme activity.

### 3.6. The Effect of Temperature on the Activity of Immobilized Enzyme

To study the effect of temperature on free enzymes and immobilized enzymes, the free enzymes and immobilized enzymes were put in different temperature reaction systems to catalyze glucose, the enzyme activities under different conditions were recorded, and the highest activity of the free enzymes was defined as 100% relative enzyme activity.

### 3.7. POD-Like Activity and Kinetic Assay of FeS_2_ Nanozyme

TMB was used as the substrate for the measurement of peroxidase-like (POD-like) activity with H_2_O_2_. Different concentrations of H_2_O_2_ and FeS_2_ were mixed in a phosphate buffer with pH = 4.5, and 10 μL of 80 mM TMB solution was then added. The absorbance of the mixture at 652 nm was assessed using a UV-vis spectrophotometer (Perkin Elmer, Waltham, MA, USA) after 5 min of reaction at 37 °C [42].

The nanozyme’s affinity for its substrate was assessed by measuring the Michaelis constant (*K_M_*) defined as the substrate concentration at which the nanozyme exhibits half the maximum reaction rate. Maximum reaction velocity (*V_max_*) was defined as the maximum reaction rate measured at a saturated substrate concentration. The kinetic constants *K_M_* and *V_max_* were determined using Origin software, by fitting the substrate concentrations and initial reaction velocities (*v*) to the Michaelis-Menten equation:*v* = (*V_max_* × [S])/(*K_M_* + [S])(1)
where [S] is the substrate concentration. The initial reaction velocity was calculated using the equation:*v* = △A/(△t × ε × L)(2)
where △A is the value of the change in absorbance before and after the reaction; △t (s) is the reaction time; ε (M^−1^ cm^−1^) = 39000 is the molar absorption coefficient of the colorimetric substrate, and L (cm) is the width of the quartz cuvette [32].

### 3.8. Enzyme Activity Assay of Gox

The reducing sugar assay was used to measure the amount of glucose consumed. Briefly, 100 μL of the Gox reaction mixture containing residual glucose was mixed with 200 μL of DNS reagent and heated at 100 °C for 10 min. Then 900 μL of distilled water was added to the mixture, and the absorbance at 540 nm was measured. The consumption of glucose was used to calculate the enzyme activity titer of Gox.

### 3.9. Immobilized Enzyme Cycle Assay

In short, the enzyme activity of Gox in the FeS_2_/CS/Gox system was determined first, and the FeS_2_/CS/Gox system was filtered and dried after the reaction was complete. For subsequent enzymatic activity assays, the FeS_2_/CS/Gox system obtained above was used. Enzyme activity was recorded for each reaction.

### 3.10. Construction of the System for Glucose Detection

The system for glucose detection consisted of 1mg of FeS_2_/CS@Gox, 890 μL of 0.1 M phosphate buffer (pH = 4.5), and 10 μL of TMB (80 mM). The amount of 100 μL of the sample to be tested was then added to the glucose detection system. After 5 min of reaction at room temperature, the absorbance at 652 nm was measured, and the glucose concentration was determined using a standard curve. The LOD was calculated using the equation:
LOD = 3·σ/*s*(3)
where σ is the standard deviation of 20 measurements, and *s* is the slope of the standard curve.

### 3.11. The Selectivity of FeS_2_ for H_2_O_2_ Detection and Glucose Detection

The selectivity of FeS_2_ for H_2_O_2_ was detected in solutions containing H_2_O_2_ (1 mM) or disrupters (1 mM). These assays were tested in the solution which contained TMB (1mM) and FeS_2_ (100 µg mL^−1^). Color change of the above solutions was observed at 2 min, and their absorbance was recorded at 652 nm. Pure HAc-NaAc buffer was used as a control.

The selectivity of the FeS_2_/CS/Gox system for glucose was detected in a solution containing glucose (50 μM) or disrupters (50 μM). The amount of 100 µL of glucose solution (100 µM) or other interfering chemicals (100 µM) was mixed with 100 µL of 1 mg mL^−1^ FeS_2_/CS/Gox system. Finally, a color change of the above solutions was observed, and their absorbance was recorded at 652 nm.

### 3.12. Preparation of Urine Simulant

For the preparation of the urine-simulated solution, a certain amount of glucose was dissolved in water first, and then the same amount of amino acids and urea as the glucose was added. After thorough mixing, impurities were filtered out and stored at 4 °C for future use. For the urine-simulated solution, the glucose content is known, and the FeS_2_/CS/Gox system is used to measure the glucose content in the solution and compare it with the actual content.

### 3.13. Shelf Life of the Glucose Detection System Assay

The effectively created FeS_2_/CS@Gox system was dispersed in an aqueous solution initially, and then it was kept in an aqueous solution for various amounts of time to assess its stability. Glucose was detected by the Gox system, and the stability of the FeS_2_/CS@Gox system was assessed using absorbance at 652 nm.

## 4. Conclusions

In this study, we used FeS_2_ nanozymes to encapsulate Gox, and the resulting immobilized enzyme system demonstrated increased catalytic efficiency and resistance to harsh conditions. FeS_2_/CS@Gox was further used to construct a glucose detection system with great selectivity, sensitivity, and accuracy which has the potential to be applied to the actual detection of glucose in urine and blood. This system demonstrates that nanozymes may be efficient carriers of natural enzymes and that the cascade interaction between nanozymes and natural enzymes is a powerful strategy for advancing the use of natural enzymes. It is anticipated that the biocatalytic catalyst nanozyme will serve as a link between biological and abiotic catalysis.

## Figures and Tables

**Figure 1 molecules-28-01357-f001:**
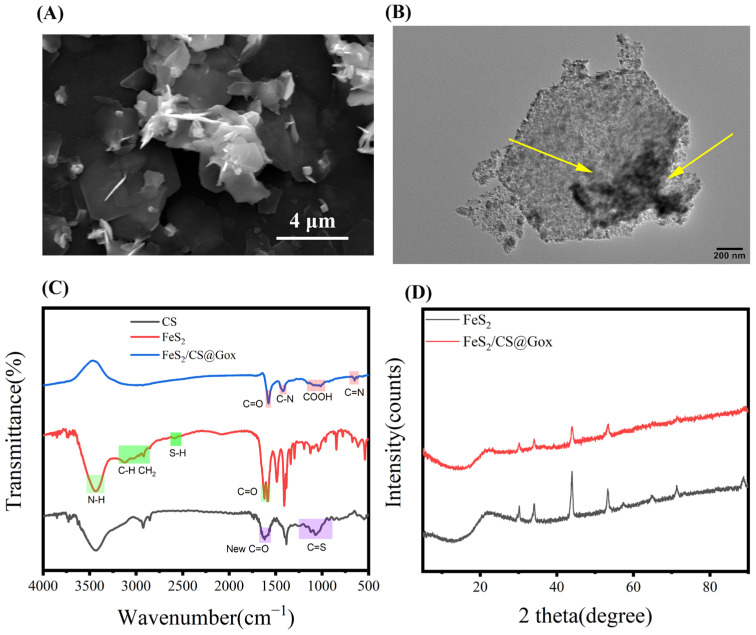
(**A**) The SEM picture of FeS_2_/CS@Gox. (**B**) The TEM picture of FeS_2_/CS@Gox (The yellow arrow points to the attachment on FeS_2_). (**C**) The FTIR of CS, FeS_2_ and FeS_2_/CS@Gox. (**D**) The XRD pattern of FeS_2_ and FeS_2_/CS@Gox.

**Figure 2 molecules-28-01357-f002:**
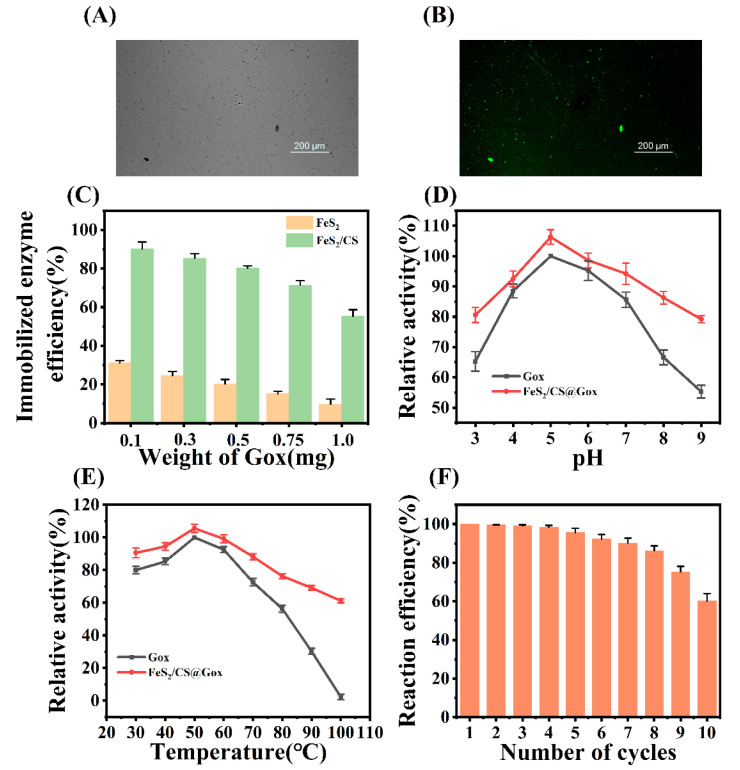
(**A**) Image of the immobilized enzyme under a confocal laser microscope. (**B**) Green fluorescence of the immobilized enzyme. (**C**) Immobilized enzyme efficiency of FeS_2_ and FeS_2_/CS (**D**) Effect of pH on Catalytic Efficiency of Immobilized Enzymes. (**E**) The Effect of Temperature on the Catalytic Efficiency of Immobilized Enzymes. (**F**) Reuse Efficiency of Immobilized Enzymes.

**Figure 3 molecules-28-01357-f003:**
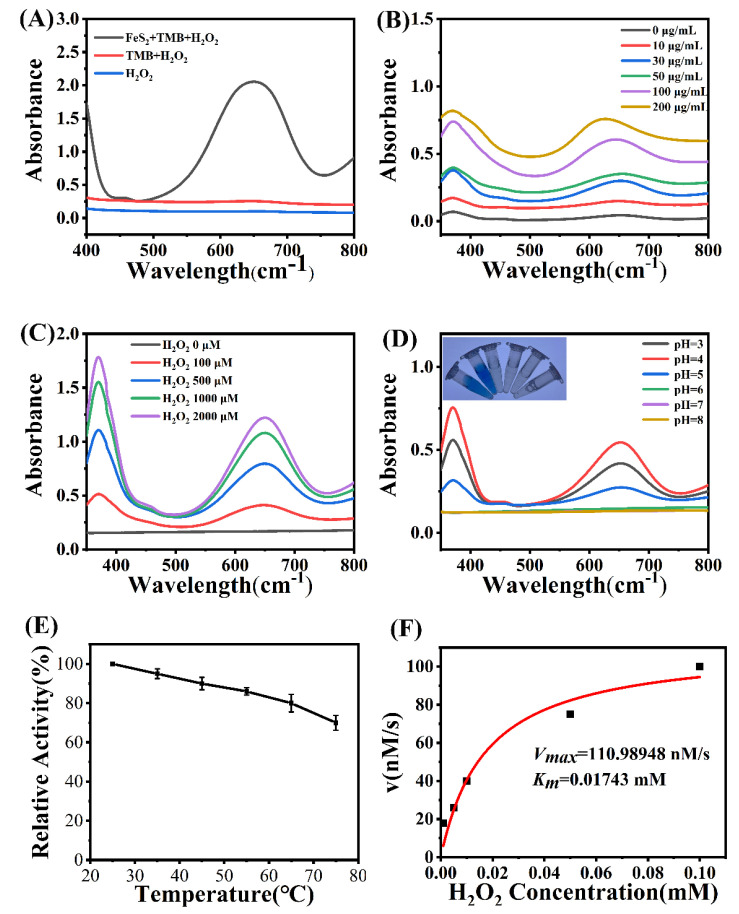
(**A**) UV−vis absorbance spectra changes of TMB in different reaction systems. (**B**) UV−vis absorbance spectra changes of TMB in different concentrations of FeS_2_. (**C**,**D**) UV−vis absorbance spectra changes of TMB in different concentrations of H_2_O_2._ (**E**) The effect of temperature on the catalytic activity of POD-like FeS_2_. (**F**) Kinetic assay for the POD-like activity of FeS_2_ nanozymes with H_2_O_2_ as substrate.

**Figure 4 molecules-28-01357-f004:**
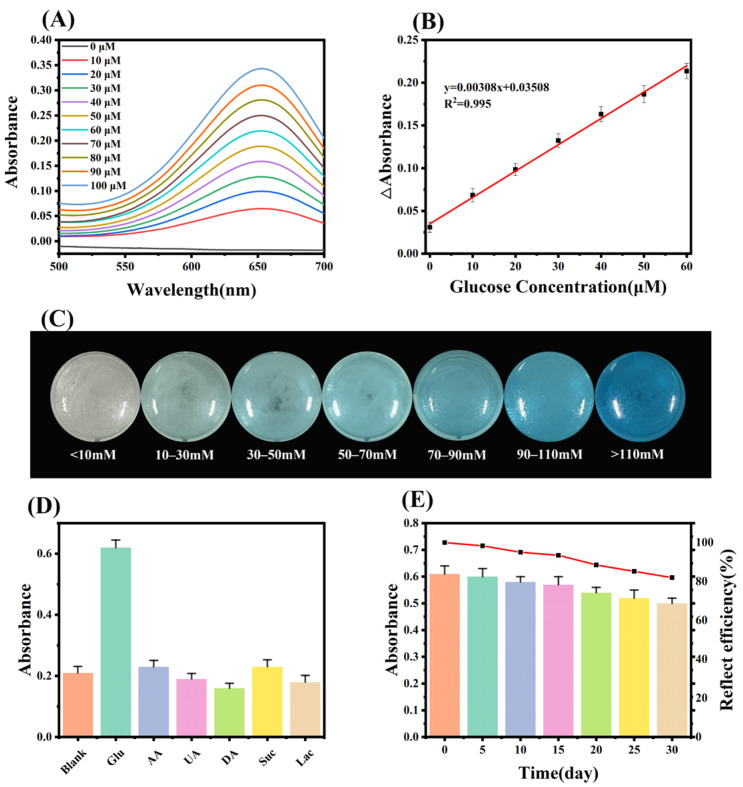
(**A**) UV–vis absorption spectra of FeS_2_/CS@Gox in different glucose concentrations. (**B**) Linear calibration curve of the absorbance at 652 nm against glucose concentration. (**C**) Standard color chart of glucose concentration and color change of FeS_2_/CS@Gox. (**D**) Selectivity toward glucose of the above system against potential interfering substances. (**E**) Shelf life of the glucose detection system.

**Figure 5 molecules-28-01357-f005:**
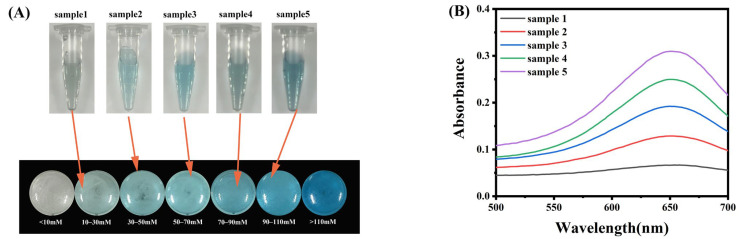
(**A**) Actual detection of color changes for different samples. (**B**) UV–vis absorbance spectra changes for different samples.

**Table 1 molecules-28-01357-t001:** Comparison of the colorimetric glucose sensors using various artificial peroxidase.

Material	Bioassay Type	Linear Range	LOD(μM)	Ref.
FeS_2_	Solution	0–60 μM	1.9 μM	This work
Fe SSN	Solution	0–60 μM	2.1 μM	[44]
Fe_3_O_4_	Solution	62.5–500 μM	50 μM	[45]
CoO-OMC	Solution	0–500 mM	68 μM	[46]
Au-Ag-Pt	Solution	0–10 mM	289.6 μM	[47]

**Table 2 molecules-28-01357-t002:** The actual detection effect of different samples.

Sample	Abs	Con (μM)	Actual Con (μM)	Error (%)
Sample 1	0.06669	10.263	10	2.63
Sample 2	0.12885	30.445	30	1.48
Sample 3	0.19211	50.984	50	1.97
Sample 4	0.24967	69.672	70	0.05
Sample 5	0.30984	89.208	90	0.88

## Data Availability

Not applicable.

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
