# Peer review of "Dual-Enzyme Cascade Composed of Chitosan Coated FeS2 Nanozyme and Glucose Oxidase for Sensitive Glucose Detection"

_molecules, 2023, doi:10.3390/molecules28031357_

Round 1

Reviewer 1 Report

In the article Dual-enzyme cascade composed of chitosan coated FeS2 nanozyme and glucose oxidase for sensitive glucose detection, Bowen Shen, Molan Qing, Liying Zhu, Yuxian Wang, Ling Jiang, the FeS2 nanocomposite was used in combination with glucose oxidase for glucose determination. Various techniques were used to characterize the FeS2 nanocomposite, mainly UV Vis in experiments with H2O2 and glucose.

The topic after the number of overviews placed in the Introduction is very current, but after reading the article I did not notice a clearly formulated aspect of novelty. Is this aspect of the novelty a new material, a new enzyme, the method of its immobilization on the material, durability of the system, the cost of materials, or maybe the lowest detection limit described so far in the literature for interferents, or maybe determination in urine is a novelty here. For this, please include a table in which the results will be compared to other systems, for example LOD and others.

Please also refer to this work: Sensitive colorimetric assay of hydrogen peroxide and glucose in humoral samples based on the enhanced peroxidase/mimetic activity of NH2-MIL-88-derived FeS2@CN nanocomposites compared to its precursors, Ru Fan, Jinrong Tian, Huili Wang, Xuedong Wang, Peipei Zhou, Microchimica Acta (2022) 189:427, https://doi.org/10.1007/s00604-022-05525-w

If this system described in the work is better than those described in the literature, it is worth publishing this work in the journal.

Author Response

尊敬的编辑/审稿人,

感谢您对我们手稿“壳聚糖包被的FeS2纳米酶和葡萄糖氧化酶组成的用于灵敏葡萄糖检测的双酶级联反应”(分子-2155627)的有用意见和建议。这些意见对我们的论文的修改和完善都很有价值,对我们的研究具有重要的指导意义。这次修改后,我们已经对稿件进行了相应的修改,修改后的稿件中突出了我们稿件的变化。此外,我们已经写了一封逐点回复信,如下所示。

编辑和审稿人评论:审稿人#1:

在壳聚糖包被的FeS2纳米酶和葡萄糖氧化酶组成的双酶级联反应用于灵敏的葡萄糖检测中,沈博文,清莫兰,朱丽颖,王玉贤,蒋玲,将FeS2纳米复合物与葡萄糖氧化酶联合用于葡萄糖测定。在H2O2和葡萄糖的实验中,使用各种技术来表征FeS2纳米复合材料,主要是紫外可见光。

引言中列出的概述数量之后的主题非常最新,但是在阅读了这篇文章之后,我没有注意到新颖性的一个明确表述的方面。新颖性的这一方面是一种新材料,一种新的酶,其固定在材料上的方法,系统的耐用性,材料的成本,或者可能是迄今为止文献中描述的干扰物的最低检测限,或者也许在尿液中的测定在这里是一种新颖性。为此,请附上一个表格,其中将结果与其他系统(例如 LOD 和其他系统)进行比较。

另请参阅这项工作:基于NH2-MIL-88衍生的FeS2@CN纳米复合材料与其前体相比增强的过氧化物酶/模拟活性的体液样品中过氧化氢和葡萄糖的灵敏比色测定,樊如,田金荣,王慧丽,王学东,周佩佩,微化学学报(2022)189:427,https://doi.org/10.1007/s00604-022-05525-w

如果作品中描述的这个系统比文献中描述的系统更好,那么值得在期刊上发表这项工作

响应:感谢您提出这个问题。在这项工作中,我们想揭示纳米酶可以作为生物酶的优良载体,因此检测葡萄糖的应用诞生了。我相信我们的FeS 2是一种优秀的纳米酶,但由于FeS2在检测葡萄糖之前用于酶固定化,因此它的LOD并不比本文更好(Microchimica Acta (2022) 189:427),但与其他工作相比,我们建立的系统也具有明显的优势。我们列入了一个表格以进一步澄清。

T能够 1.比较使用各种人工过氧化物酶的比色葡萄糖传感器。

材料

生物测定类型

线性范围

LOD(微米)

裁判。

二氧化

溶液

0-60微米

1.9微米

这项工作

铁 SSN

溶液

0-60微米

2.1微米

45

3O4

溶液

62.5-500微米

50微米

46

CoO-OMC

溶液

0-500 毫米

68微米

47

金-银-铂

溶液

0-10 毫米

289.6微米

48

第 225 行,第 8 页。

我们还认为基于纳米酶的传感系统具有更多优势,并参考本文(微化学学报 (2022) 189:427)。

第 214 行,第 7 页。参考文献44.

Reviewer 2 Report

The article «Dual-enzyme cascade composed of chitosan coated FeS2  nanozyme and glucose oxidase for sensitive glucose detection» has a high applied value. The text is written in a logical and accessible way. The scheme of the experiment is quite understandable and reproducible.

Authors demonstrates that nanozymes may be employed as ideal carriers for biological enzymes, and that the nanozymes can catalyze cascade reactions together with natural enzymes, offering new insights into interactions between natural and synthetic biosystems. In this study, glucose oxidase was covalently bound to the FeS2 nanozyme as the immobilization carrier, while chitosan encapsulation increased the activity and stability of the immobilized enzyme. The immobilized enzyme exhibited a 10% increase of catalytic efficiency than over free enzyme, while also being more stable and catalytically active in environments with an alkaline pH of 9.0 and a high temperature of 100℃.

Minor remarks:

1. Line 92, in designation “KM” m should be lowercase

2. Line 138, preposition “of” written twice

3. Figure 4 and the caption to it should be placed on one page.

In general, the article makes a good impression. I recommend to publish it

Author Response

Dear Editor/ Reviewers,

Thank you for your useful comments and suggestions of our manuscript “Dual-enzyme cascade composed of chitosan coated FeS2 nanozyme and glucose oxidase for sensitive glucose detection” (molecules-2155627). Those comments are all valuable and very helpful for revising and improving our paper, as well as the important guiding significance to our researches. After this revision, we have already modified the manuscript accordingly, and the changes of our manuscript were highlighted in revised manuscript. Besides, we have written a point-by-point response letter as you can see below.

Editor and Reviewer comments:
Reviewer #2:

The article «Dual-enzyme cascade composed of chitosan coated FeS2 nanozyme and glucose oxidase for sensitive glucose detection» has a high applied value. The text is written in a logical and accessible way. The scheme of the experiment is quite understandable and reproducible.

Authors demonstrates that nanozymes may be employed as ideal carriers for biological enzymes, and that the nanozymes can catalyze cascade reactions together with natural enzymes, offering new insights into interactions between natural and synthetic biosystems. In this study, glucose oxidase was covalently bound to the FeS2 nanozyme as the immobilization carrier, while chitosan encapsulation increased the activity and stability of the immobilized enzyme. The immobilized enzyme exhibited a 10% increase of catalytic efficiency than over free enzyme, while also being more stable and catalytically active in environments with an alkaline pH of 9.0 and a high temperature of 100℃.

Minor remarks:

  1. Line 92, in designation “KM” m should be lowercase

Response: Thank you for raising this problem. We have corrected KM in the text. Line 92.

  1. Line 138, preposition “of” written twice

Response: Thank you for your careful reading of this manuscript. This was caused by our own carelessness, we have removed the redundant “of “.

  1. Figure 4 and the caption to it should be placed on one page.

Response: Thank you for your recognition of our work. For the overall beauty of the manuscript, we put Fig 4 and Caption on the same page.

Round 2

Reviewer 1 Report

I have no more comments. I accept the article in the current version.